# Cost-effectiveness of mandatory folic acid fortification of flours in prevention of neural tube defects: A systematic review

Viviane Belini Rodrigues[ID]*, Everton Nunes da Silva, Maria Leonor Pacheco Santos

Graduate Program Collective Health, University of Brasilia, Brasilia, District federal, Brazil

* vivianebelini@gmail.com

**Data Availability Statement:** All relevant data are within the manuscript and its Supporting Information files.

## Abstract

### Background

Neural tube defects (NTDs) constitute the most frequent group among congenital malformations and are the main cause of neonatal morbimortality. Folic acid (FA) can reduce the risk of pregnancies affected by NTDs.

### Objective

We aimed to investigate whether mandatory folic acid (FA) fortification of flours is cost-effective as compared to non-mandatory fortification, and to verify whether FA dosage, cost composition, and the quality of economic studies influence the cost-effectiveness of outcomes.

### Methods

We conducted a systematic review. The protocol was registered on PROSPERO (CRD 42018115682). A search was conducted using the electronic databases MEDLINE/ PubMed, Web of Science, Embase, Scopus, and EBSCO/CINAHL between January 2019 and October 2020 and updated in February 2021. Eligible studies comprised original economic analyses of mandatory FA fortification of wheat and corn flours (maize flours) compared to strategies of non-mandatory fortification in flours and/or use of FA supplements for NTD prevention. The Drummond verification list was used for quality analysis.

### Results

A total of 7,859 studies were identified, of which 13 were selected. Most (77%; $n = 10$) studies originated from high-income countries, while three (23%) were from upper-middle-income countries. Results of a cost-effectiveness analysis showed that fortification is cost-effective for NTD prevention, except for in one study in New Zealand. The cost-benefit analysis yielded a median ratio of 17.5:1 (0.98:1 to 417.1:1), meaning that for each monetary unit spent in the program, there would be a return of 17.5 monetary units. Even in the most unfavorable case of mandatory fortification, the investment in the program would virtually payoff at a ratio of 1:0.98. All FA dosages were cost-effective and offered positive health gains, except in one study. The outcomes of two studies showed that FA dosages above

**Funding:** The author(s) received no specific funding for this work.

**Competing interests:** The authors have declared that no competing interests exist.

300 µg/100 g have a higher CBA ratio. The studies with the inclusion of "loss of consumer choice" in the analysis may alter the fortification cost-efficacy ratio.

## Conclusion

We expect the findings to be useful for public agencies in different countries in decision-making on the implementation and/or continuity of FA fortification as a public policy in NTD prevention.

## Introduction

Neural tube defects (NTDs) constitute the most frequent congenital malformations and are the main cause of neonatal morbimortality [1]. Anencephaly, encephalocele, and spina bifida are the most frequent alterations of the central nervous system and result from the incomplete closure of structures that originate in the brain and the spinal cord between the third and fourth weeks after conception [2]. NTDs can lead to death or varying degrees of disability. The etiology remains unknown; however, there is an interaction of several factors, such as nutritional, environmental, and genetic [3]. Folic acid (FA) can reduce the risk of pregnancies affected by NTDs [4]. A prior review [5] presented evidence that FA consumption in the periconception period prevents NTDs occurrence or recurrence. Research shows that women of childbearing age do not obtain the daily FA recommendation (0.4 mg) from dietary foods alone [6,7].

There are 81 countries that adopt mandatory FA fortification in foods [8], and 71 countries with immediate potential for mandatory fortification of 145 million tons of flour with FA. Thus, approximately 57,000 live births associated with spina bifida and anencephaly would be preventable annually, resulting in global prevention from 13% to 34% [9]. In 1998, the United States and Canada adopted mandatory fortification, and 10 years later verified NTDs reduction of 28% and 40%, respectively [10,11]. Chile implemented mandatory fortification of flour in 2000, and 2 years later, there was a 40% reduction in the prevalence of NTDs [12]. At the end of 2002, in Brazil, the National Sanitary Surveillance Agency (ANVISA) decided that FA fortification of wheat and corn flours (also known as maize flours) should become mandatory from June 2004 [13]. The country registered a reduction in NTDs prevalence of 30%, from 0.79/1000 in the pre-fortification period (2001–2004) to 0.55/1000 in the post-fortification period (2005–2014) [14].

A recent systematic review [15] demonstrated that in countries with mandatory fortification the estimate of spina bifida prevalence was lower among live births, stillbirths, and pregnancy interruptions (35.22/100,000 live births) when compared to voluntary fortification (52.29/100,000 live births). Fortification of flour with FA is more effective in preventing NTDs because it does not require changes in eating habits and ensures that women of childbearing age have access to FA in the periconception period [16].

A systematic review [17] showed the high economic burden of NTDs in health systems and societies, concluding that FA interventions are cost-effective. Although there is evidence in favor of mandatory fortification with FA, there is lack of evidence about which FA dosage is more cost-effective. This aspect is particularly relevant in the context of high heterogeneity in FA dosage among mandatory fortification programs across the world. There is also lack of evidence about the cost composition and quality of economic evaluations conducted on mandatory fortification with FA.

While the last review on economic assessment was published 10 years ago (2011), this updated systematic review is justified by the need to investigate the cost-effectiveness of NTD prevention strategies in low- and high-income economies, and to verify whether FA dosage, cost composition, and the quality of economic studies influence the cost-effectiveness of outcomes. The differential of this review is the detailed methodological characterization and systematization of economic models, the selection of original economic assessments, and the inclusion of recent economic evaluations on the theme.

In this context, we aimed to investigate whether mandatory FA fortification of flours is cost-effective as compared to non-mandatory fortification, and to verify whether FA dosage, cost composition, and the quality of economic studies influence the cost-effectiveness of outcomes.

## Materials and methods

### Protocol and register

This systematic review was conducted following international [18] and national [19] recommendations. The report followed the recommendations of PRISMA [20] (see the S1 File). The protocol was registered on PROSPERO (CRD 42018115682). https://www.crd.york.ac.uk/PROSPEROFILES/115682_STRATEGY_20210326.pdf.

### Eligibility criteria

Eligible studies were original economic analyses on the strategy of mandatory FA fortification of wheat and corn flours (maize flours) compared to those of non-mandatory FA fortification of flours and/or use of FA supplement for NTDs prevention. There were no restrictions on the studies' language and year of publication.

This review excluded qualitative studies, expert opinions, letters to editors, book chapters, editorials, abstracts, conference paper, biofortification studies, notes, literature review, and any other type of critical essay or studies without original economic analysis.

### Data and research sources

Searches were conducted using the electronic databases MEDLINE/PubMed, Web of Science, Scopus, Embase, EBSCO/CINAHL, between January 2019 and October 2020 updated in February 2021. The search strategy was applied on MEDLINE and used to search other databases (see the S2 File). The strategy contains both MeSH terms and keywords commonly used in the field of economic studies and food fortification. The reference lists of the included studies were manually reviewed to identify additional publications.

### Selection of studies and data collection process

The selection of studies was conducted by two independent and blinded reviewers (VBR and DSP). The reviewers analyzed the titles and abstracts of studies identified in the databases that contemplated the research question. The studies that complied with the inclusion criteria were integrally read (full text). The reviewers read independently, and any dissent was resolved consensually; in cases with no consensus, the issue was resolved by a third reviewer (ENS).

Extracted data included the authorship, country, year of publication, currency, year of cost collection, type of economic assessment, the study perspective, time horizon, description of assessed strategies (intervention versus comparer), health outcomes (NTD reduction, avoided deaths, avoided costs, avoided disability-adjusted life years, years of life lost and quality-adjusted life years), costs (direct, indirect, intangible), type of disease, analytical model,

discount rate, sensibility analysis, incremental cost-effectiveness ratio (ICER), and conclusion of the study.

To provide a measure of comparable outcomes between the selected studies, data were extracted on the investment return of programs of mandatory FA fortification of wheat and corn flours, measured by the ratio between the monetary benefit of program implementation (avoided costs) and the total cost of the program. Thus, a measure of return for every dollar invested in the mandatory fortification program is presented. When this measure was unavailable, calculations were made based on data reported in the study.

## Quality assessment

For quality assessment of economic studies, the reduced verification list of Drummond and colleagues was used [21]. The verification list contained 10 items that enabled the critical assessment of studies regarding the following criteria: 1) well-defined research question; 2) comprehensive description of concurrent alternatives; 3) efficacy evidence of established programs/services; 4) relevance of costs and consequences; 5) precision measure of costs and consequences; 6) credibility of cost values and consequences; 7) time adjustment of costs and consequences; 8) use of incremental cost analysis and consequences of alternatives; 9) sensibility analysis; and 10) adequate discussion (based on index or calculation, comparison of outcomes with similar studies, discussion of generalization of outcomes, evaluation of factors, and implementation issues). The fulfillment of these criteria enabled the measurement of the quality of the studies. The following cut-off points were used: high-quality study (8–10 items fulfilled), medium-high quality study (6–7 items fulfilled), medium-low quality study (4–5 items fulfilled), and low-quality study (<3 or less items fulfilled) [22].

## Data synthesis

The main methodological characteristics, quality of the included studies and their respectively results were summarized in tables. We also provided a comparable measure of the economic benefit of the mandatory FA fortification, by means of the return of the investment. When the return of investment was not available in the included studies, we calculated it by dividing total cost averted attributed to mandatory FA fortification by the total cost of the mandatory FA fortification program. It means that for each monetary unit spent on the program, there would be a return of x monetary units.

**Heterogeneity between studies.** The heterogeneity observed between the selected studies was analyzed by making subgroups such as FA dosage, cost composition, income difference per country, and differences in the methodological quality of the studies.

## Results

### Description of eligible studies

The search strategy used on the eight databases enabled the identification of 7,859 studies, of which 13 studies fulfilled the eligibility criteria. The flow diagram outlines the article selection process as shown in Fig 1.

The studies were published during 1995–2006 [23–26], 2007–2017 [23–34], and in 2019 [35]; they were mostly (77%; *n* = 10) published in countries with high-income economies [23,25,26,29–35], while and three studies [24,27,28] in those with upper-middle-income economies. No study has been conducted in low-income country.

Some studies [25–28,30–35] used the difference in NTD estimates between the pre-post fortification collection baseline data for birth defects surveillance systems or hospital sentinels

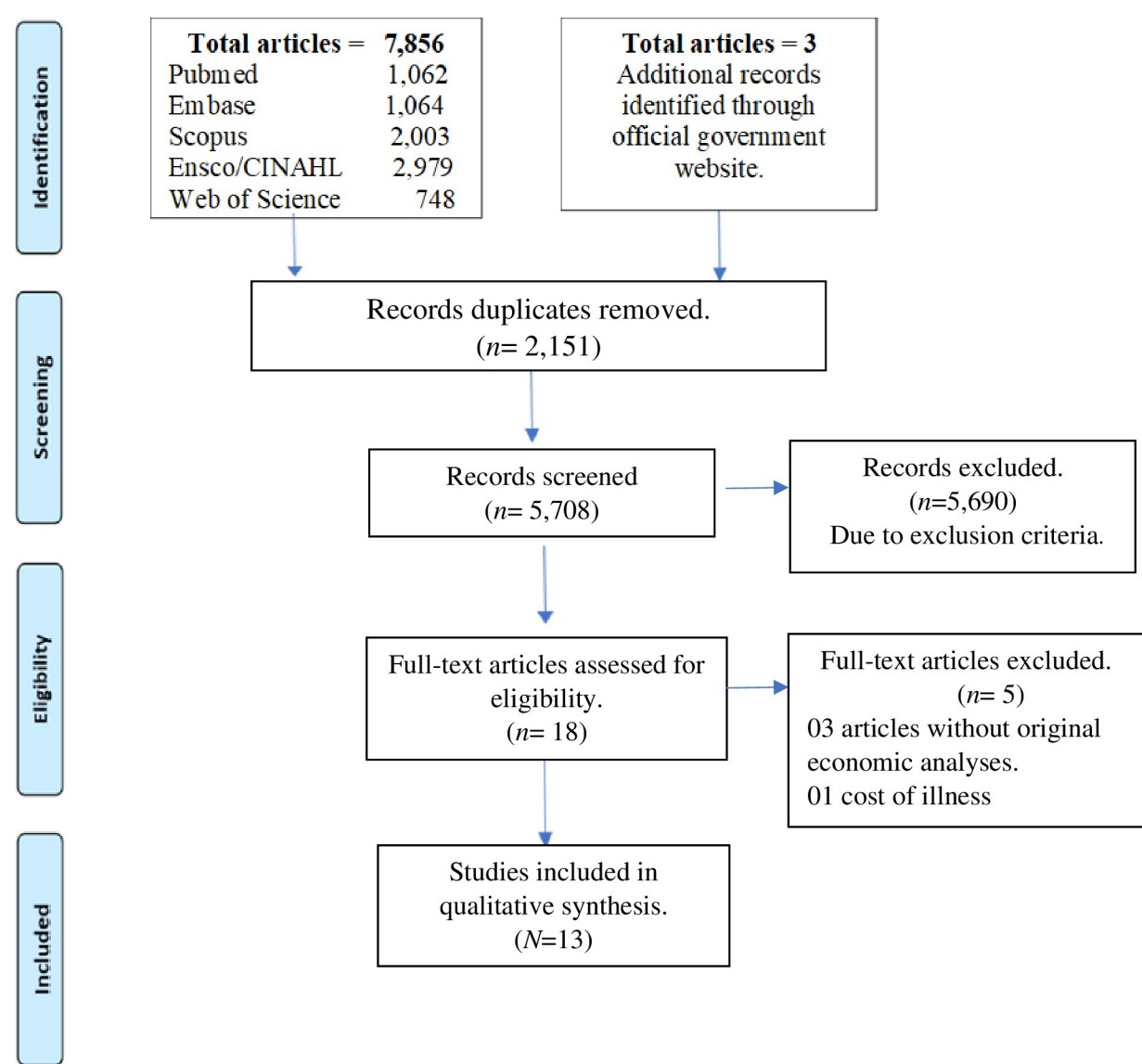

**Fig 1. Flow diagram of article selection process.**

[28]. Three studies [23,24,29] used estimates of NTD reduction from the U.S. The adopted perspective was that of society [24,26,29,32–35]; however, four studies did not inform a perspective and in two, it was not possible to identify the perspective [27,31]. The time horizon was the expected lifetime for persons with spina bifida and encephalocele [29,30,32–35], and three studies [24,26,31] adopted a period of 10 to 15 years. The others did not inform a time horizon [23,25,27] or it was not possible to identify it [28].

For the economic analysis, there was a combination of two or more types, such as effectiveness (CEA) and cost-utility analyses (CUA) [25,27,29,31,32,35] and cost-benefit (CBA) and cost-utility (CUA) and cost-effectiveness analyses [34]. However, some studies presented only one type of analysis, such as CUA [30] or CBA [23,24,26,28,33]. In these studies, deterministic univariate sensitivity analysis was conducted for the main outcomes and costs, except for one study [28], which did not provide this information.

The adverse effect of neuropathy was observed in only three studies [23,30,32]. Bentley and colleagues [30], included the adverse effect (neuropathy) stratified by sex and age group. Benefits exceeded costs with fortification, despite estimates of collateral effects in all age groups and race/ethnicity in the population (Tables 1 and 2).

**Heterogeneity in FA dosage.** The economic models presented a wide variability in FA quantities, varying from 100 μg to 700 μg per 100 g of flour. The most frequent quantities were 140 μg and 200 μg folic acid per 100 g of flour [23,25–28,31,32,35], owing to recommendations from each country's regulatory agency. Only one study [30] compared the economic and health outcomes of fortification levels of 140 μg, 300 μg, and 700 μg per 100 g of flour. A previous study [24] did not report the FA quantity.

The outcomes of studies [23,30] showed that FA dosages above 300 μg/100 g have a higher cost-benefit ratio to FA dosage inferior to 300 μg/100 g. Notably, all FA dosages were cost-effective and offered positive health gains, except in one study [32].

**Heterogeneity in the composition of costs.** Several studies [23,26,27,31,32,34] described direct costs for the private sector and public administration. For milling industries, the studies attributed costs of production, storage, and product distribution to the market. The regulatory costs included national education campaigns, and inspection of enriched products. Four studies have included the costs of surveillance of adverse effects on the population [27,32,34,35].

Although the studies in this review presented cost-effective outcomes with fortification in NTD prevention, only six studies showed that fortification was also cost-saving [25,27–30,33]. It is observed in these studies that the direct costs refer only to the fortification process and NTD treatment, except for two studies [29,30] that included non-medical costs (home layout adequacy and special education). Other studies [23,24,26,31,32,34,35] added different variables to direct costs, such as national education/awareness campaigns for the target population and training of professionals, which were cost-effective, but not cost-saving, which can be attributed to substantial spending on education campaigns at the national level.

Two [31,32] also highlighted that the inclusion of "loss of consumer choice" in the analysis may alter the fortification cost-efficacy ratio.

Regarding indirect costs, most studies [25,27,28,30,32,33] did not report the composition. The others [23,25,26,29,33–35] included calculation variables such as caregiver time (hours), reduction in the labor force and productivity, reposition time of labor force (employer), payment of pensions, and other social benefits to persons with NTDs.

The most frequent non-medical cost composition in the studies [23,25,26,29–33] were home adequacy, special education, and the time (hours) caregivers dedicated to patients. Only one study [29] included travel costs for parents during patients' hospitalization (Table 3).

**Heterogeneity in ex-ante and ex-post economic assessments of fortification.** Economic studies were conducted at different times: some before [23,24,26] and others during or after fortification implementation [25,27–35]. Depending on the time when the assessment was conducted, countries will present different outcomes. This is owing to the difference between the *ex-ante* analysis, which aims to estimate the feasibility of the program before its implementation, and the *ex-post* assessment, which presents the actual performance of the policy on the population [36].

Grosse and colleagues [25] highlighted that *ex-ante* evaluation underestimates the effects of fortification in NTD reduction (~2%) when compared to the outcomes obtained in the *ex-post* assessment (~30%). The authors attributed the difference to the consumption of products with higher FA doses and the unknowingness of the dose-response curve of the vitamin.

In this review, some *ex-ante* evaluations [29,30,32] have estimated the avoided costs with neuropathies (adverse effect) even without robust scientific evidence between FA dosage added to flours (~200 μg) and the effect on health. Consequently, the outcomes of these economic evaluations could be underestimated.

**Table 1. Methodological characterization of selected studies.**

| Frist Author | Country | Year of publication | Currency | Year of cost | Type of Study | Perspective | Time horizon (year) | Mandatory Fortification (acid folic/ flour) | Comparison strategies | Health Outcomes |
|---|---|---|---|---|---|---|---|---|---|---|
| Romano [23] (Ex-ante) | U.S. | 1995 | U.S. dollar | 1991 | CBA | Not reported | Not reported | 140 mcg/100 g 350 mcg/100 g | Dietary supplement. | NTD avoided cases |
| Bagriansky [24] (Ex-ante) | Kazakhstan | 2003 | U.S. dollar | Not reported | CBA | Society | 10 years | Not reported | Non-fortification | Net benefit |
| Grosse [25] (Ex-post) | U.S. | 2005 | U.S. dollar | 2002 | CEA/CUA | Not reported | Not reported | 140 mcg | Non-fortification | Avoided NTD births and net benefit |
| FSANZ [26] (Ex-ante) | Australia | 2006 | U.S. dollar | 2005 | CBA | Society | 15 years | 100 mcg/100 g 200 mcg/100 g | Voluntary fortification | Net benefit (avoided DALYs) Avoided costs |
| | New Zealand | | New Zealand dollar | | | | | | | |
| Llanos [27] (Ex-post) | Chile | 2007 | International dollar | 2001 | CEA/CUA | Not reported | Not reported | 200 mcg/100 g | Non-fortification | Avoided fetal deaths Avoided DALY Avoided NTDs |
| Jentink [28] (Ex-ante) | Netherlands | 2008 | Euro | 2005 | CEA/CUA | Society | Life-long | 140 mcg/100 g | Non-fortification Voluntary fortification | QALY, YLL |
| Sayed [29] (Ex-post) | South Africa | 2008 | Rand (ZAR) | Not reported | CBA | Not reported | Not reported | Wheat flour 1.5 mg/kg Corn flour 2.21 mg/kg | Non-fortification | Avoided NTD costs |
| Bentley [30] (Ex-post) | U.S. | 2009 | U.S. dollar | 2005 | CUA | Not reported | Life-long | 140 mcg/100 g 300 mcg/100 g 700 mcg/100 g | Non-fortification | QALY |
| Dalziel [31] (Ex-ante) | Australia | 2009 | Australian dollar | 2006 | CEA/CUA | Not reported | CEA 10 years CUA Life expectancy 80 years | 200 mcg/100 g 135 mcg/100 g | National program promoting: 1) dietary supplement use. 2) voluntary fortification extension. 3) campaign to increase consumption of FA-rich unprocessed food. | Avoided NTD cases Avoided DALY |
| | New Zealand | | New Zealand dollar | | | | | | | |
| Rabovskaja [32] (Ex-ante) | Australia | 2013 | Australian dollar | 2005 | CEA/CUA | Society | Life-long | Bread flour: 200 mcg/100 g | Voluntary fortification | QALY years of life; avoided NTD cases |
| Grosse [33] (ex-post) | U.S. | 2016 | U.S. dollar | 2014 | CBA | Society | Life-long | 140 mcg/100 g | NTDs prevalence pre- and post- fortification | Net benefit |
| FSANZ [34] (Ex-post) | Australia | 2017 | Australian dollar | 2014 | CBA/CEA/CUA | Society | Lifetime 82.3 years (maximum) | Bread flour: 200 mcg/100 g | Non-mandatory fortification | Avoided NTD cases; years of life; QALY |

*(Continued)*

**Table 1.** (Continued)

| Frist Author | Country | Year of publication | Currency | Year of cost | Type of Study | Perspective | Time horizon (year) | Mandatory Fortification (acid folic/ flour) | Comparison strategies | Health Outcomes |
|---|---|---|---|---|---|---|---|---|---|---|
| Saing.[35] (Ex-post) | Australia | 2019 | Australian dollar | 2014 | CEA/ CUA | Society | Lifetime 78 years | 200 mcg/100 g | Voluntary FA fortification of flours (including education and supplementation programs). | QALY; years of life; avoided NTD cases. |

CEA (cost-effectiveness analysis); CUA (cost-utility analysis); CBA (cost-benefit analysis); NTD (neural tube defect); QALY (quality-adjusted life years); DALY (disability-adjusted life years); YLL (Years of life lost).

**Heterogeneity in high-income and upper-middle-income countries.** The outcomes of the cost-effectiveness analysis conducted in high-income countries [23,25–27,29,30,33–35] showed fortification as cost-effective in NTDs prevention, except for New Zealand [32]. In this case, the results of the analysis showed that the benefits do not surpass the costs of fortification of 100 μg/ 100 g in bread flour, which may be related to the lower reduction rate (~8%) of NTDs cases.

In upper-middle-income countries [24,28], the fortification was cost-effective; however, in one study [28] conducted in South Africa, it appeared to be cost-saving.

**Measures of return for each dollar invested in mandatory fortification program.** Relating the potential benefits of mandatory fortification implementation (avoided costs) to its respective costs (program costs) resulted in a median ratio of 17.1:1 (variation of 0.98:1 to 417.1:1), which means that for each monetary unit spent on the program, there would be a return of 17.5 monetary units. Even in the most unfavorable case to mandatory fortification [32], the investment in the program would virtually pay for itself, considering that there would be a ratio of 1 (cost of the program) to 0.98 (avoided costs). This broad interval can be attributed to the difference in the composition of costs with fortification or NTD treatment used in economic models among countries (Table 4).

We emphasize that the wide difference between the mean 68.7 and the median 17.5 is owing to the CBA ratio estimates presented in the study by Bentley and colleagues [30]. The economic model estimated the costs of mandatory fortification for three levels of FA (140, 350 and 700 mcg/100 g) and the benefits for four diseases: NTD; myocardial infarction (MI); colon cancer (CC); and vitamin B12 masking by sex, age, and race/ethnicity (non-Hispanic white/ black and Mexican American. Because it was not possible to disaggregate costs by disease, we used median and ratio measures for comparison with other studies.

## Quality of economic evaluations

Regarding the quality of the13 selected studies, only one study [34] fulfilled all criteria of the Drummond tool [21]. However, five studies [26,31–33,35] contemplated eight to nine items of the verification list, with high-quality data. The others were classified as medium-high-quality [25,29] and low-quality [23,24,27,32] because they fulfilled four to seven criteria of the list. Only one study [28] presented fewer than 3 fulfilled items, thus representing studies with low-quality data (see the S3 File).

## Discussion

The reduced number of original economic assessments on the theme of this review (*n* = 13) highlights a field with scarce research in the face of a problem of such social and economic

**Table 2. General characterization of costs of selected studies.**

| Author | Disease conditions | Analytical models | Discount rate | Sensibility analysis | Results |
|---|---|---|---|---|---|
| Romano [23] | Spina bifida Anencephaly Neuropathy | Not reported | 2.5% and 6% | Deterministic univariate | Net economic benefits 4.3:1 (140mcg) and 6.1:1 (350 mcg) |
| Bagriansky [24] | Spina bifida Anencephaly Heart disease | Not reported | 5% | Deterministic univariate | 10-Year Benefit Ratio is 11.7 with an Internal Rate of Return estimated at 319%. |
| Grosse [25] | Spina bifida Anencephaly | Not reported | 3% | Deterministic univariate | The cost savings (net reduction in direct costs) were estimated to be in the range of $88 million to $145 million per year. |
| FSANZ [26] | Spina bifida Anencephaly and Encephalocele | Not reported | 3.3% and 3.8% | Deterministic Analysis | In both Australia and New Zealand, the benefits of mandatory fortification of all bread making flour with folic acid outweigh the costs. |
| Llanos [27] | Spina bifida Anencephaly | Not reported | 3% | Deterministic univariate | Considering averted costs of care, fortification resulted in net cost savings of I$ 2.3 million |
| Jentink [28] | Spina bifida | Decision tree | 1.5% to 4% | Deterministic univariate | Our model suggests that AF fortification of bulk food to prevent cases of NTD in newborns might be a cost-saving intervention in the Netherlands. |
| Sayed [29] | Orofacial clefts and Spina bifida. | Not reported | Not reported | Not reported | The cost benefit ratio in averting NTDs was 46 to 1. |
| Bentley [30] | Spina bifida, Anencephaly Heart attack, Colon cancer Masking of Vit B12 deficiency | Markov | 3% | Deterministic univariate. | Compared with no fortification, all post-fortification strategies provided QALY gains and cost savings for all subgroups. |
| Dalziel [31] | Spina bifida Encephalocele | Not reported | 5% | Deterministic univariate | Mandatory fortification was not cost-effective for New Zealand at $AU 138,500 per DALY ($US 109,609, £56,216), with results uncertain for Australia, given widely varying cost estimates. |
| Rabovskaja [32] | Spina bifida Anencephaly Neuropathies | Decision tree | 5% | Deterministic univariate | Mandatory fortification was cost-effective at A$10,723 per LYG and at A$11,485 per QALY. However, inclusion of the loss of consumer choice can change this result. |
| Grosse [33] | Spina bifida Anencephaly | Not reported | Not reported | Deterministic, per scenario | Fortification with folic acid is effective in preventing NTDs and saves hundreds of millions of dollars each year. |
| FSANZ [34] | Spina bifida Anencephaly Encephalocele | Decision tree | Not reported | Deterministic univariate | Mandatory fortification was cost-effective, equitable, and efficient in comparison with the set of pre-mandatory fortification policies. |
| Saing [35] | Spina bifida Anencephaly Encephalocele | Decision tree | 5% | Deterministic univariate probabilistic (Monte Carlo simulation). | Mandatory folic acid fortification (in addition to policies including advice on supplementation and education) improved equity in certain populations and was effective and highly cost-effective for the Australian population. |

NTD (neural tube defect); FA (folic acid); ICER (incremental cost-effectiveness ratio); LYG (life years gained); DALY (disability-adjusted life years); QALY (quality-adjusted life years).

magnitude that affects all countries. Majority of studies were from three countries, the U.S., Australia, and New Zealand, classified by the World Bank as high-income countries. Nevertheless, all selected economic studies showed that the economic burden of NTDs is considerable, and that FA fortification engendered economic and health benefits, except for one study that reported it as not being cost-effective in all scenarios, especially by including the cost of "loss of consumer choice."

Segal and colleagues [37] suggest that the loss of consumer choice should be considered as a cost for the population, even when there is another option of flour (fortifier-free) because the choice will remain restricted. The method consisted of attributing a value in U.S. dollars (US$ 1) to this loss, affecting thousands of people who do not belong to the target population of the intervention, substantially increasing the cost of mandatory fortification.

**Table 3. Types of costs included in the studies selected in our systematic review.**

| First author | Medical costs | | | | | | | No medical costs | | | Indirect costs | | |
|---|---|---|---|---|---|---|---|---|---|---|---|---|---|
| | Mandatory fortification | Regulatory costs | Inpatient care | Outpatient care | Drugs | Exams | Side effects (fortification) | Assistance technology | Transport, care holder | Home adequacy, special education | Absenteeism | Presenteeism | Premature death |
| Romano [23] | Yes | No | Yes | Yes | Yes | Yes | Yes | No | No | Yes | No | No | Yes |
| Bagriansky [24] | Yes | Yes | Yes | Yes | No | No | No | No | No | No | No | No | Yes |
| Grosse [25] | Yes | No | Yes | Yes | No | No | No | No | No | Yes | No | No | No |
| FSANZ [26] | Yes | No | Yes | Yes | Yes | Yes | No | Yes | No | Yes | Yes | No | Yes |
| Llanos [27] | Yes | No | Yes | Yes | No | No | No | No | No | No | No | No | No |
| Jentink [29] | Yes | No | Yes | Yes | No | Yes | No | Yes | Yes | Yes | Yes | No | No |
| Sayed [28] | Yes | No | Yes | Yes | Yes | Yes | No | No | No | No | No | No | No |
| Bentley [30] | Yes | No | Yes | Yes | Yes | Yes | Yes | No | No | Yes | No | No | No |
| Dalziel [31] | Yes | Yes | Yes | Yes | Yes | Yes | No | No | No | No | Yes | No | No |
| Rabovskaja [32] | Yes | Yes | Yes | Yes | Yes | Yes | Yes | Yes | No | Yes | No | No | No |
| Grosse [33] | Yes | No | Yes | Yes | Yes | No | No | No | No | Yes | No | No | No |
| FSANZ [34] | Yes | Yes | Yes | Yes | Yes | No | Yes | Yes | No | No | Yes | No | No |
| Saing [35] | Yes | No | Yes | Yes | No | No | No | Yes | No | No | Yes | Yes | No |

**Table 4. Return of the investment on mandatory FA fortification of the included studies.**

| Author/year | Frequency | Folic acid (100 mg/ product) | Benefit (currency) | Cost (currency) | Return of investment |
|---|---|---|---|---|---|
| Romano [23] | 4.6:10 000 | 140 mcg | US$ 121,500,000.00 | US $ 27,940,000.00 | 4,3:1 |
| | | 350 mcg | US$ 300,900,000.00 | US $ 49,200,000.00 | 6,1:1 |
| Bagrianski [24] | 73:10 000 | Not reported | US$ 21,747.90 | US $ 1,454.00 | 14,9:1 |
| Grosse [25] | Not reported | 140 mcg | US$ 145,000,000.00 | US $ 3,000,000.00 | 48,3:1 |
| FSANZ [26] | 72 cases incidents | 100 mcg 200 mcg | NZ$ 21,593,681.00 NZ$ 43,521,621.00 | NZ$ 2,336,910.00 NZ$ 2,348,658.00 | 9,2:1 18,5:1 |
| | 338 cases incidents | 100 mcg 200 mcg | A$ 50,285,100.00 A$ 125,703,672.00 | A$ 1,152,357.00 A$ 1,208,357.00 | 43,6:1 104,0:1 |
| Ilanos [27] | 14.8:10 000 | 150 mcg | I$ 2,300,000.00 | I$ 208,000.00 | 11,5:1 |
| Sayed [28] | 1.4:1000 | Not reported | R 28,456,946.00 | R 1,400,000.00 | 20,3:1 |
| Jentink [29] | 9.0:10 000 | 140 mcg | € 1,802,836.00 | € 686,000.00 | 2,6:1 |
| Bentley [30] | 10.6:10 000 | 140 mcg | US$ 783,750,000.00 | US$ 3,300,000.00 | 237,5:1 |
| | | 300 mcg | US$ 2,534,700,000.00 | US$ 6,000,000.00 | 422,5:1 |
| | | 700 mcg | US$ 4,592,700,000.00 | US$ 10,500,000.00 | 417,1:1 |
| Dalziel [31] | 1.43:1000 Not reported | 200 mcg low-cost scenario | A$ 2,008,720.00 | A$ 26,995,208.00 | 13,4:1 |
| | | 200 mcg high-cost scenario | A$ 14,694,120.00 | A$ 262,291,544.00 | 17,8:1 |
| | | 135 mcg | NZ$ 3,819,840.00 | NZ$ 79,524,528.00 | 20,8:1 |
| Rabovskaja [32] | 13.3:10 000 | 200 mcg | A$ 5,886,630.00 | A$ 5,780,423.00 | 0,98:1 |
| Grosse [33] | 6.5:10 000 | 140 mcg | US$ 791,900,000.00 | US$ 20,000,000.00 | 25,2:1 |
| FSANZ [35] | 10.2:10 000 | 200 mcg | A$ 1,471,717,219.00 | A$ 1,472,371,542.00 | 1,0:1 |
| Saing [36] | 10.2:10 000 | 200 mcg | A$ 2,054,765.00 | A$ 604,439.00 | 3,4:1 |

The literature has also showed that higher dosage of FA is associated with higher reduction of NTD cases. For example, an increase in FA intake of 200 mcg/day would reduce the risk if NTD by 23% in Britain, while an increase of 400 mcg/day would reduce it risk by 36% [38]. There is also evidence against setting any upper intake level of FA, including FA dosage higher than 1000 mcg/day would increase the opportunity to prevent NTDs worldwide [39,40]. Our findings are in line with these results from the literature, since higher FA dosage were associated with better value for money and higher return of the investment from mandatory FA fortification programs.

It was not possible to conduct a comparison of outcomes in the economic assessments owing to differences in the measures of benefits, perspectives, time horizon, and currency.

The variability in cost composition, calculation methodology, and discounts in the studies can be attributed to the differences between the pharmacoeconomic guidelines of the countries [40]. A previous study [25] reported difficulty in obtaining precise indirect costs, which may have contributed to its non-inclusion in past economic assessments. Hence, costs regarding

caregiver time and reduction of the labor force, among other indirect costs, were inferred, and when they are incorporated into the economic studies, it may alter the cost-saving ratio.

In general, the studies were of medium to low quality, although some used international methodological patterns in economic assessments [41]. Some were based on data from observational studies, which may jeopardize the reliability of outcomes. However, the studies published in the last 6 years presented a substantial improvement in the report quality, and most of the population databases were from the post-fortification stage.

The present study adds new evidence that makes mandatory FA fortification of wheat/maize flours a very cost-effective and potentially cost-saving strategy in different social, economic, and health system settings. The Global Nutrition Report [42] estimated that every dollar invested in nutrition intervention yields $16 in health and productivity benefits. We demonstrated that the return of investment of mandatory fortification exceeded this estimate, provided an average return of 17,5:1 for every dollar invested in the program.

The contribution of mandatory FA fortification to the global economy by reducing the infant mortality rate and expenditures for health systems [17,43] is clear. Over 90% of low-income countries have high mortality rates among 28-day old infants owing to pneumonia, diarrhea, and congenital defects [44]. Because of insufficient healthcare coverage, flour fortification provides a timely and cost-effective option to avoid neonatal mortality from congenital defects, such as NTDs. In high-income countries, due to the insufficient FA ingestion by women of the reproductive age (10–49 years) [45–47], mandatory fortification is an efficient strategy to avoid NTDs.

## Limitations

Some limitations of this study must be acknowledged. First, only studies of high- and medium-high-income countries were identified. Therefore, outcomes cannot be extrapolated to low- and medium-income economy countries, although these have higher potential to obtain significant gains from mandatory FA fortification, given the low coverage of healthcare services and the high proportion of the population living in socioeconomic vulnerability conditions. Second, the outcomes were analyzed qualitatively since comparison between studies is not possible owing to methodological and healthcare system differences. Hence, we propose a comparable measure of investment returns for future research in this area. Third, the interventions varied between studies, reflecting the diversity of local regulations on mandatory fortification programs in terms of FA dosage. Therefore, we performed a subgroup analysis.

## Implications for healthcare policies and systems

Although the results obtained from our systematic review are from high and upper-middle income countries, nutritional deficiencies and their consequences occur at different levels among countries, particularly in low- and middle-income countries [48].

According to the 2020 Global nutrition Report [49], there is a need to establish synergy between public health and equity through strategies that improve the nutritional status of the entire population. This will reduce healthcare costs and save lives. Moreover, the Sustainable Development Goals (SDG) 2030 Agenda [50] is aligned with the focus on reducing inequalities among populations worldwide.

Mandatory fortification provides equity in different health systems worldwide. Prior studies [34,35] affirm that fortification improved equity compared to the pre-mandatory fortification policy (voluntary fortification and FA supplement). Further, a Brazilian study [14] revealed a decreased prevalence of NTDs by 37.7% among the offspring of adolescent mothers.

We understand that FA fortification does not replace the use of FA supplementation and much less the actions of health education professionals. These strategies, when combined, potentiate the prevention of NTDs. The WHO and FAO recommend food fortification, drug supplementation, and nutrition education to increase FA intake among women of childbearing age [51]. Therefore, the role of the academic community is crucial for the improvement of fortification and the assessment of the impact in different economic contexts and healthcare systems. Finally, it is hoped that the scientific evidence of this study will subsidize public agents of different countries in favor of the implementation and/or continuity of FA fortification of flours as a public policy intervention for the prevention of NTDs.

## Supporting information

**S1 File. PRISMA checklist.**
(DOC)

**S2 File. Search strategy.**
(DOCX)

**S3 File. Quality of studies.**
(DOCX)

## Author Contributions

**Conceptualization:** Viviane Belini Rodrigues, Everton Nunes da Silva, Maria Leonor Pacheco Santos.

**Data curation:** Viviane Belini Rodrigues, Everton Nunes da Silva, Maria Leonor Pacheco Santos.

**Formal analysis:** Viviane Belini Rodrigues, Everton Nunes da Silva, Maria Leonor Pacheco Santos.

**Investigation:** Viviane Belini Rodrigues, Everton Nunes da Silva.

**Methodology:** Viviane Belini Rodrigues, Everton Nunes da Silva.

**Supervision:** Maria Leonor Pacheco Santos.

**Writing – original draft:** Viviane Belini Rodrigues, Everton Nunes da Silva.

**Writing – review & editing:** Viviane Belini Rodrigues, Everton Nunes da Silva, Maria Leonor Pacheco Santos.

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
