## [Decision Letter · Decision Letter 0]

10 Jun 2021

PONE-D-21-10351

Cost-effectiveness of mandatory folic acid fortification of flours in prevention of neural tube defects: a systematic review

PLOS ONE

Dear Dr. Rodrigues,

Thank you for submitting your manuscript to PLOS ONE. After being reviewed by 2 experts in the field of food fortification, we feel that your manuscript has merit but does not fully meet PLOS ONE’s publication criteria as it currently stands. Therefore, we invite you to submit a revised version of the manuscript that addresses the points raised during the review process.

Please pay attention to the remarks of the first reviewer to better put forward the purpose of your review, and to put it better in the current global context.

We look forward to receiving your revised manuscript.

Kind regards,

Frank Wieringa, M.D., Ph.D.

Academic Editor

PLOS ONE

Journal Requirements:

Additional Editor Comments (if provided):

Reviewers' comments:

Reviewer's Responses to Questions

**Comments to the Author**

1. Is the manuscript technically sound, and do the data support the conclusions?

Reviewer #1: Yes

Reviewer #2: Yes

2. Has the statistical analysis been performed appropriately and rigorously? 

Reviewer #1: Yes

Reviewer #2: Yes

3. Have the authors made all data underlying the findings in their manuscript fully available?

Reviewer #1: Yes

Reviewer #2: Yes

4. Is the manuscript presented in an intelligible fashion and written in standard English?

Reviewer #1: No

Reviewer #2: Yes

5. Review Comments to the Author

Reviewer #1: In this systematic review, the authors review and summarize the cost-effectiveness of mandatory folic acid fortification of flours for preventing neural tube defects while additionally aiming to verify whether the folic acid dosage, cost compositions and quality of the economic studies included influences the cost effectiveness of the outcomes. 13 studies were identified that provide information enabling the authors to conclude that folic acid fortification is cost-effective in all but one of the countries included. A comparison between high and low-income countries was not feasible as no studies were identified in low-income countries. The findings of this study will be useful to promote the implementation of folic acid fortification to prevent neural tube defects globally in places where it is not currently implemented.

Major comments:

The introduction does not sufficiently articulate the context and rationale for this study. The authors should rewrite the introduction to better describe the current global context of folic acid fortification and what is known about its cost-effectiveness and what evidence gaps remain that this review will fill. While the final paragraph provides some specific reasons justifying for the review, these do not all appear to be supported by the preceding background information and its justification should be strengthened.

The protocol appears to be technically sound; however, the presentation of the results (particularly Tables 1 and 2) could be improved to improve understanding and readability. Currently, there is some overlap in content (e.g. comparison strategies appears in both) and table 2 is very long with varying levels of details across the studies. The authors should review and revise the tables such that they provide a more concise summary of only the most important characteristics and include a similar level of detail across all studies.

The discussion should be revised to better articulate the overall findings of the review (as they relate to the aims of the paper, including not only the cost-effectiveness of folic acid fortification but also the verification of the influencing factors that were assessed) and the implications for folic acid fortification to prevent NTDs globally. In addition, a comparison to what is already known on cost-effectiveness appears to be missing and is important given past systematic reviews have been done on the subject (as mentioned in the introduction) and this would help to put these results in context and support the conclusions.

Finally, the conclusion raises research gaps that are not directly related to the study at hand. This should be rewritten to more strongly to summarize the findings, relevant research gaps, and what this means for the global community.

Minor comments:

1. The manuscript would benefit from an overall review by a copy editor as some phrases are awkwardly worded and hard to understand.

2. Corn flour is often referred to as maize flour in many countries; therefore, it would be helpful to note that somewhere in the manuscript if in fact they are the same.

3. The addition of a background statement at the start of the abstract before describing the aim of the study would be useful to provide context.

4. The aim of the study in both the abstract and introduction should be revised to be more clearly and consistently worded as what is written in lines 84-85 varies from what is stated earlier in line 21-23 and line 79 about wanting to compare low vs. high income economies and now stating the comparison is mandatory vs. non mandatory fortification programs.

5. The findings on factors that influence the cost-effectiveness of folic acid fortification should be added to the abstract as this was an aim of the study.

6. Line 85-86: this statement does not add anything and could be deleted.

7. Lines 144-145: the reference to table 3 should be in results, not methods.

8. Lines 166-168: sentence is not clearly written - what is meant by “epidemiologic database” and what “other countries” are referred to?

9. Lines 174-175: abbreviations CEA/CUA/CBA need to be written out at first mention in the text.

10. In table 1: Use of the response “not clear” should be revised to “not included or does not report” where possible as this is not typically a standard assessment.

11. In table 2: The final column “Author’s conclusions” should be renamed “Results” and should be a brief summary with similar level of detail across studies.

12. Table 1 & 2: NI is not a standard abbreviation.

13. Table 3: Folic acid contents column is repetitive of what is in Table 1

14. Table 4: This should be made supplementary material and the results briefly summarized in the text.

15. Line 253: results for “mediated ratio of 18.5:1” - no information is provided on how this was calculated. if it is an average of all country results then details on this should be added to the methods section.

Reviewer #2: The study examined the cost-effectiveness of mandatory fortification of flours, compared to non-mandatory fortification, with folic acid through a systematic review of literature. Authors have identified all key papers on the topic and presented an adequate analysis answering their research question. The manuscript is well-written and presents policy makers important knowledge on fortification effectiveness while planning birth defects prevention programs in their country. Best and conservative scenarios of cost benefit presented in this paper also help in making a strong case for fortification.

Comments to Authors:

Page 3: Lines 61-68. I do not think this paragraph adds to the paper. I suggest you either delete it or bring it up in the Discussion with following focus: The evidence on safety of folic acid is well established by many previous papers, and summarized very well by Field and Stover which you have already cited (Reference: Field MS, Stover PJ. Safety of folic acid. Ann N Y Acad Sci. 2018 Feb;1414(1):59-71. doi: 10.1111/nyas.13499). Presenting something that is not definitive distracts from your work. There is also no basis for upper limit for folic acid as written by Nick Wald and can be used instead of the information presented (Ref: Wald NJ, Morris JK, Blakemore C. Public health failure in the prevention of neural tube defects: time to abandon the tolerable upper intake level of folate. Public Health Rev. 2018 Jan 31;39:2).

Can you add a recommendation for cost-benefit analysis for mandatory fortification much needed in low-income countries as this will be key in promoting fortification policies in high burden countries.

6. PLOS authors have the option to publish the peer review history of their article (what does this mean?). If published, this will include your full peer review and any attached files.

Reviewer #1: No

Reviewer #2: No

---

## [Author Response · Author response to Decision Letter 0]

8 Aug 2021

Please, the attached file "response to reviewers".

---

## [Editor Report · Decision Letter 1]

29 Sep 2021

Cost-effectiveness of mandatory folic acid fortification of flours in prevention of neural tube defects: a systematic review

PONE-D-21-10351R1

Dear Dr. Rodrigues,

We’re pleased to inform you that your manuscript has been judged scientifically suitable for publication and will be formally accepted for publication once it meets all outstanding technical requirements.

Kind regards,

Frank Wieringa, M.D., Ph.D.

Academic Editor

PLOS ONE
---

## [Editor Report · Acceptance letter]

4 Oct 2021

PONE-D-21-10351R1 

Cost-effectiveness of mandatory folic acid fortification of flours in prevention of neural tube defects: a systematic review 

Dear Dr. Rodrigues:

I'm pleased to inform you that your manuscript has been deemed suitable for publication in PLOS ONE. Congratulations! Your manuscript is now with our production department. 

Kind regards, 

on behalf of

Dr. Frank Wieringa 

Academic Editor

PLOS ONE